# Reduction of LPAR1 Expression in Neuroblastoma Promotes Tumor Cell Migration

**DOI:** 10.3390/cancers14143346

**Published:** 2022-07-09

**Authors:** Xiangjun Liu, Mengmiao Pei, Yongbo Yu, Xiaolin Wang, Jingang Gui

**Affiliations:** 1Laboratory of Tumor Immunology, Beijing Pediatric Research Institute, Beijing Children’s Hospital, Capital Medical University, National Center for Children’s Health (NCCH), Beijing 100045, China; somusl@126.com (X.L.); mengmiao_pei@163.com (M.P.); 2Beijing Key Laboratory for Pediatric Diseases of Otolaryngology, Head and Neck Surgery, Beijing Pediatric Research Institute, Beijing Children’s Hospital, Capital Medical University, National Center for Children’s Health (NCCH), Beijing 100045, China; yuyongbo1688@126.com

**Keywords:** LPAR1, neuroblastoma, LPA, tumor metastasis, bioinformatics analysis

## Abstract

**Simple Summary:**

The tumor metastasis in the bone marrow or other organs in high-risk neuroblastoma patients is a serious problem to tackle and strongly impairs the survival of patients. Novel and effective targets for the treatment of neuroblastoma, especially tumor metastasis, need to be explored. Using multiple databases and analysis methods, LPAR1 was screened out through our comprehensive bioinformatics analysis and found to be positively associated with survival of neuroblastoma patients. LPAR1 was proved to be reduced in neuroblastoma cells compared with non-mailgant cells. LPA-LPAR1 axis showed migration-inhibitory effects on neuroblastoma cells, suggesting that LPAR1 may be a potential target for future treatment of neuroblastoma.

**Abstract:**

Neuroblastoma is the most common extracranial solid tumor in children. Tumor metastasis in high-risk NB patients is an essential problem that impairs the survival of patients. In this study, we aimed to use a comprehensive bioinformatics analysis to identify differentially expressed genes between NB and control cells, and to explore novel prognostic markers or treatment targets in tumors. In this way, *FN1*, *PIK3R5*, *LPAR6* and *LPAR1* were screened out via KEGG, GO and PPI network analysis, and we verified the expression and function of LPAR1 experimentally. Our research verified the decreased expression of LPAR1 in NB cells, and the tumor migration inhibitory effects of LPA on NB cells via LPAR1. Moreover, knockdown of LPAR1 promoted NB cell migration and abolished the migration-inhibitory effects mediated by LPA-LPAR1. The tumor-suppressing effects of the LPA-LPAR1 axis suggest that LPAR1 might be a potential target for future treatment of NB.

## 1. Introduction

Neuroblastoma (NB), an embryonic tumor of the sympathetic nervous system that arises in the fetus or early after birth from sympathetic cells produced by the neural crest, is a significant cause of childhood death [1]. Although some NBs automatically degenerate and have a good prognosis, tumor metastasis in the bone marrow or other organs in high-risk NB patients is still an essential problem to tackle [2,3]. It is necessary to find novel, effective targets for the treatment of NB, especially tumor metastasis. Microarray technology and bioinformatics analysis are increasingly used to explore the significant genetic or epigenetic variations in tumors and determine cancer diagnoses and prognoses, as well as determine treatment targets [4]. In this study, we aimed to identify novel diagnostic biomarkers or therapeutic targets and determine the pathogenesis in NB using bioinformatics analysis and experimental verification.

The Gene Expression Omnibus (GEO) is an international public knowledge base for archiving and distributing microarrays, next-generation sequences and other forms of high-throughput functional genomic data free of charge [5]. We accessed publicly available data on NB cells and non-malignant control cells and screened differentially expressed genes (DEGs) subjected to Kyoto Encyclopedia of Genes and Genomes (KEGG) and Gene Ontology (GO) analyses. The Protein-Protein Interaction (PPI) network and Molecular Complex Detection (MCODE) plug-in were further employed to boil down the critical DEGs including *FN1*, *PIK3R5*, *LPAR6* and *LPAR1*. Through an investigation into the expression patterns and potential regulating functions of the screened genes in NB cells, our research mainly focused on exploring the expression and function of LPAR1.

LPAR1 is a member of the G protein-coupled receptor family of lysophosphatidic acid (LPA) receptors (LPARs), including LPAR1 to LPAR6 [6]. LPA is a small phospholipid generally found in serum, ascitic effusions and inflammatory fluids [7]. LPA acts as an extracellular signaling molecule by binding to and activating its receptors LPARs, thereby exerting regulating functions in cellular proliferation/migration/survival, vascular homeostasis, stromal remodeling, lymphocyte trafficking and immune regulation [8,9,10]. Aberrant LPAR1 expression was evident in a variety of cancer cell lines and primary tumors [6]. LPAR1 was significantly downregulated in prostate cancer, and high LPAR1 expression was correlated with a favorable overall survival [11]. Furthermore, LPAR1 was reported to mediate migration- or invasion-inhibiting signals in prostate cancer [7], gastric cancer [12] and pancreatic cancer [13]. In a rat neuroblastoma cell line or mouse fibroblast cell line, overexpression of LPAR1 also markedly decreased intrinsic cell motility and invasion [14,15].

Our results show the decreased expression of LPAR1 in NB cells, demonstrating that LPA can exert tumor migration-inhibitory effects on NB cells via LPAR1. Knockdown of LPAR1 also promotes NB cell migration and abolishes the migration-inhibitory effects mediated by LPA-LPAR1.

## 2. Materials and Methods

### 2.1. Microarray Data Collection and Preprocessing

We searched the microarray gene expression datasets associated with neuroblastoma from GEO (https://www.ncbi.nlm.nih.gov/geo/ (accessed on 27 November 2021) for the study. GEO, NCBI’s publicly available genomics database, which collects submitted high-throughput gene expression data, was thoroughly queried for all datasets involving studies on NB. Datasets were related to a neuroblastoma group and a negative control group in humans. Our inclusion criteria were as follows: (1) Expression profiling data by microarray; (3) Complete microarray normalized data. Ultimately, we chose six NB cells from GSE28019, GSE16477 and GSE90804 and three non-malignant control cells from GSE10592, GSE24733 and GSE57864, using the raw data in our study. The data were based on the [HG-U133_Plus_2] Affymetrix Human Genome U133 Plus 2.0 Array Plate. First, the dataset was quality-controlled before differential genetic screening analysis, which included use of the “affyPLM” package in R software (R version 4.0.4) to verify the conformance of parallel trials. Then, a robust multi-array averaging (RMA) algorithm was applied using the “affy” package in R to convert the raw array of data into expression values and to perform background correction, normalization and probe summarization [16,17]. Both a *p*-value < 0.01 and log2 fold change >2.1 were considered critical for DEG screening based on the paired *t*-test of the “limma” R package [18].

### 2.2. Functional and Pathway Enrichment Analysis of Neuroblastoma-Specific DEGs

GO is a community-based bioinformatics resource that supplies information about gene product functions, using ontologies to represent biological knowledge [19], thereby informing us what kinds of biological functions genes have. It mainly consists of three categories: cell composition (CC), molecular function (MF) and biological process (BP). KEGG is a knowledge base for the systematic analysis of gene functioning, linking genomic information with higher-order functional information. Genomic information is stored in the Gene Database, a collection of gene catalogs of all sequenced genomes and partial genomes, with updated annotations of gene functions [20]. GO and KEGG analyses can be found in the DAVID database (DAVID version 6.8; https://david.ncifcrf.gov/ (accessed on 27 November 2021)), which is a fully functional annotation tool providing a comprehensive set of functional annotation tools for investigators to use to understand the biological meaning behind a long list of genes [21]. A *p*-value < 0.05 was taken as the critical value when identifying DEGs using official gene symbols.

### 2.3. PPI Networks

An online biological database STRING (https://string-db.org (accessed on 27 November 2021), Version 11.0), from which we obtained information on protein co-expression, is well known for supporting protein co-expression prediction based on known and predicted gene PPI networks for the analysis of functional interactions between proteins [22]. In this work, PPI networks of co-expressed genes were established using the STRING database, and we considered interaction with a joint score >0.4 to be statistically significant. Then, the resulting network data were imported into local software Cytoscape (https://cytoscape.org/ (accessed on 27 November 2021), version 3.8.2) to be further visually analyzed. The functional interactions between proteins provide insights into the mechanisms of disease development, which we can access by visualizing molecular interaction networks and biological pathways and integrating these networks with annotations [23], gene expression profiles and other state data.

### 2.4. The PPI Networks and Module Selection

Clustering coefficients were calculated by the Molecular Complex Detection (MCODE) plugin in Cytoscape, and modularity was used to identify modules in the co-occurrence networks [24]. The Degree and Betweenness are factors of the topological algorithm and shortest path, respectively. We used degree cut-off = 2, node score cut-off = 0.2, k-core = 2 and max. depth = 100 as the MCODE plug-in default parameters.

### 2.5. Cell Lines

Neuroblastoma cell lines SH-SY5Y, SK-N-BE2 and IMR-32, and non-malignant cell lines RPE-1, HBE and HEK293T, were purchased from the American Type Culture Collection (Manassas, VA, USA). CHLA-255 cells were kindly provided by Prof. Shahab Asgharzadeh from the Children’s Hospital Los Angeles. SH-SY5Y, SK-N-BE2, RPE-1, HBE and HEK293T were cultured in Dulbecco’s modified Eagle’s medium (DMEM) (Corning Incorporated, Corning, NY, USA) supplemented with 10% FBS (fetal bovine serum, Gibco, Invitrogen, Carlsbad, CA, USA) and 1% penicillin/streptomycin, IMR-32 was cultured in minimum essential medium (MEM) (Corning Incorporated, Corning, NY, USA) and CHLA-255 was cultured in Iscove’s DMEM (IMEM) (Corning Incorporated, Corning, NY, USA). Cells were cultured at 37 °C in a humidified cell incubator with 5% CO_2_.

### 2.6. Transient Transfection of siRNA

Lipofectamine RNAiMAX (Invitrogen, Carlsbad, CA, USA) was used to transfect the cells with LPAR1 siRNA. To facilitate transfection, the cells were seeded to 60% confluence on a six-well plate during transfection. The next day, siRNA was transfected using RNAiMAX and Opti-MEM according to the manufacturer’s instructions. Cells were harvested or subjected to other experiments after 48 h.

### 2.7. PCR and Real-Time PCR

The total RNA from cells was isolated using TRIzol reagent (Invitrogen, Carlsbad, CA, USA). Reverse transcription was performed according to standard protocols using a RevertAid™ II First-Strand cDNA Synthesis Kit (Thermo Fisher Scientific Inc., Waltham, MA, USA). PCR and real-time PCR were performed as previously described [25]. The amplification conditions of real-time PCR were as follows: 10 min initial denaturation at 95 °C, then 40 cycles of 15 s at 95 °C and 1 min at 60 °C. The relative quantity (RQ) was calculated by the 2^ΔΔCt^ method. GAPDH was amplified as an internal standard.

The primer sequences for PCR are listed below:

LPAR1-F, 5′-AATCTATGTCAACCGCCGCT-3′

LPAR1-R, 5′-GTCAATGAGGCCCTGACGAA-3′

LPAR3-F, 5′-TTAGGGGCGTTTGTGGTATG-3′

LPAR3-R, 5′-CCTTGTAGGAGTAGATGATGGGGT-3′

LPAR6-F, 5′-CTGCGTCCTCAAAGTCCGAA-3′

LPAR6-R, 5′-CCAAATGGCCAATTCCGTGT-3′

The primer sequences for real-time PCR are listed below:

LPAR1-F, 5′-TCAACTCTGCCATGAACCCC-3′

LPAR1-R, 5′-ACTCCAGCCAAGATGGTGTG-3′

### 2.8. Cell Proliferation Assay

A Cell Counting Kit-8 (CCK8) detection kit (Dojindo Molecular Technologies, Japan) was used for measuring the cell proliferation; the cell number was directly proportional to the amount of formazan dye detected by the absorbance at 450 nm. Cells were seeded in 96-well plates at a concentration of 8000 cells per well (10,000 cells per well for CHLA-255 cells) and cultured in a complete culture medium with 10 μM LPA, LPA plus 10 μM Ki16425 or Ki16424 alone. At the indicated times, 10 μL of CCK-8 solution was added to each well. The plate was then incubated at 37 °C for 120 min, and the absorbance was detected.

### 2.9. Wound-Healing Assay

Cells were seeded in a six-well plate with 10 μM LPA, LPA plus 10 μM Ki16425 or Ki16425 alone, grown to about 80% confluence and a wound was carefully scraped with a sterilized pipette tip in the cell monolayer. After replacement with a fresh complete culture medium, photomicrographs were taken immediately, as well as 72 h after scraping. The wound widths in the pictures were measured using ImageJ software. The percentage of cell migration was calculated based on the ratio of wound width at 72 h and the initial wound width at 0 h.

### 2.10. Cell Migration Assay

Transwell chambers (8 μm pore size, BD Biosciences, NJ, USA) were used to measure cell migration. The cells were cultured in a serum-free culture medium for 12 h and then seeded in the upper chamber at a density of 1 × 10^5^ cells per well in 250 μL serum-free DMEM medium with 10 μM LPA, LPA plus 10 μM Ki16425 or Ki16425 alone. The appropriate complete culture medium was added to the lower chamber. After incubation at 37 °C with 5% CO_2_ for 24 h, the chambers’ contents were collected. The membranes were then fixed with 4% paraformaldehyde in PBS and stained with 2% crystal violet for 10 min. Photomicrographs were taken and the absolute cell numbers were counted from images captured by a microscope (100× magnification) (IX73, Olympus, Tokyo, Japan).

### 2.11. Statistical Analysis

Statistical comparisons were performed using GraphPad Prism software (version 8.0) (GraphPad Software Inc., San Diego, CA, USA). Student’s *t*-test was used to analyze the data. Error bars represented the SEM. Significant differences between groups were represented by * *p* < 0.05, ** *p* < 0.01, and *** *p* < 0.001.

## 3. Results

### 3.1. Identification of DEGs Using mRNA Microarray Data Analysis and GO/KEGG Enrichment Analysis

After collecting the mRNA microarray data of six NB cell lines (data for SH-SY5Y, SK-N-BE2 and IMR-32 cells from three different datasets) and three non-malignant cell lines (data of PRE-1, HBE and HEK293T cells from three different datasets) from the GEO database, we first performed relative log expression (RLE) boxplots analysis, and the results suggested the normalized raw data (Figure 1A). All DEGs were screened using R software (R version 4.0.4) based on an adjusted *p*-value < 0.01 and log2 fold change >2.1. Clustering analysis of these DEGs was performed using volcano plots (Figure 1B). A total of 5492 DEGs were identified from the six NB samples and the other three non-malignant control cell samples, including 38 upregulated DEGs and 5454 downregulated DEGs (Appendix A). To further our understanding of the functions of the screened DEGs, we conducted GO/KEGG enrichment analysis. All DEGs were included in the functional enrichment analysis using the DAVID database and visualized using R software. The results showed that the NB sample group had a unique GO condition. As shown in Figure 1C–E and Table 1, pathways related to extracellular matrix organization, angiogenesis, cell adhesion, positive regulation of NF-κB signaling, regulation of cell proliferation, regulation of PI3K signaling and activation of MAPK activity were enriched in GO BP analysis, and the plasma membrane, cell surface, proteinaceous extracellular matrix, cell-cell junction, focal adhesion for GO CC, calcium ion binding, receptor activity, PIK3Ca activity and cytokine receptor activity for GO MF. In terms of KEGG pathway analysis, in Figure 1F and Table 2, the NB group enriched unique pathways such as cytokine-cytokine receptor interaction, cell adhesion molecules (CAMs), the Jak-STAT signaling pathway, NF-kappa B signaling pathway, focal adhesion, the PI3K-Akt signaling pathway, pathways in cancer and the MAPK signaling pathway.

As summarized in our results, the PI3K pathway was enriched and activated in the NB group, and the PI3K pathway is generally known to activate Akt and further mediate multiple biological effects [26], including those involved in cell proliferation, apoptosis inhibition, cell migration and cell cancerous transformation, contributing much to tumorigenesis. Therefore, we subsequently screened overlapped DEGs that were statistically significant in both the PI3K-Akt signaling pathway and pathways in cancer, and 51 further screened DEGs were shown in the heatmap (Figure 1G).

### 3.2. PPI Network Construction, Module Analysis and Hub Gene Determination

PPI network analysis plays a major role in predicting the functionality of interacting genes or proteins and gives an insight into the functional relationships and evolutionary conservation of interactions among genes. Based on the screened DEGs, a PPI network was generated in the STRING protein interaction database and imported into the bioinformatics software platform Cytoscape (Version 3.8.2) for visualization and further analysis (Figure 2A). Then, the MCODE plug-in was used to select important functional modules of protein interaction networks for the identified DEGs (Figure 2B,C), and critical genes were defined according to the degree level. *FN1*, *PIK3R5*, *LPAR6* and *LPAR1* were determined to have a high degree of network connectivity. The expression levels of these four genes were shown to be decreased in NB cells (Figure 2D).

### 3.3. Hub Gene Expression and Survival Analysis

The association between hub gene expression and NB patients’ survival was analyzed using the Kaplan-Meier survival curves [27]. These were generated based on the mRNA expression levels of *FN1*, *PIK3R5*, *LPAR6* and *LPAR1,* with the log-rank test *p*-value indicated using the R2: Genomics Analysis and Visualization Platform and using Tumor Neuroblastoma-SEQC-498-custom-ag44kcwolf datasets. As shown in Figure 3A–D, survival analysis revealed that a poor prognosis was significantly associated with low *LPAR1* mRNA levels in NB patients (bonf *p* <0.05), which was the same for *FN1*, *PIK3R5* and *LPAR6*.

The results of LPAR1 expression analysis at different stages (the International Neuroblastoma Staging System (INSS)) indicated that LPAR1 showed the lowest expression level in st4 NB tumors with metastasis, rather than st4s with limited metastasis, both of which expressed lower LPAR1 levels than st1, st2 and st3 NB tumors (Figure 3E). High-risk NB tumors also showed a lower level of LPAR1 (Figure 3F). In addition, NB tumors leading to patients’ death showed significantly lower LPAR1 expression (Figure 3G), consistent with the survival curve. Expression analyses of the other three genes at different stages, risk levels and death events were performed, and the results are shown in Appendix A. PIK3R5 and LPAR6 showed similar expression patterns in NB tumors. FN1, meanwhile, demonstrated the lowest expression level in st4 NB tumors with limited metastasis, and showed no significant differences between high-risk NB tumors and NB tumors leading to patients’ death.

LPARs are the receptors of LPA and mediate the regulating function involved in multiple tumor-related cellular processes, such as proliferation/migration/survival and vascular homeostasis [8,9,10]. Our analysis suggested that both LPAR1 and LPAR6 expression were beneficial to NB patients’ survival, possibly involved in the regulation of tumor metastasis mediated by LPA.

### 3.4. NB Cells Showed Low Expression Level of LPAR1 Compared to Non-Malignant Cell Lines

According to the bioinformatics analysis results, we examined the expression of LPAR1 and LPAR6 in non-malignant cells and NB cells. The results in Appendix A show that both NB cells and non-malignant cells expressed extremely low levels of LPAR6, which made it difficult for the ligand LPA to exert functions via LPAR6. Yet, our real-time PCR and PCR results in Figure 4A,B indicate that all detected cell lines expressed LPAR1 to some extent, and NB cells, including SH-SY5Y, SK-N-BE2 and IMR-32 cells, expressed lower levels of LPAR1 compared to non-malignant cells (PRE-1, HBE and HEK293T cells). Only NB cell line CHLA-255 showed a relatively high expression of LPAR1. Therefore, we focused mainly on the expression and function of LPAR1 in NB.

### 3.5. LPA Suppressed the Migration of NB Cells via LPAR1

We had identified a relatively low LPAR1 level in NB cells and a positive correlation between LPAR1 expression and NB patient survival. To investigate the function of LPAR1 in NB, then we examined the effect of LPA, mediating intracellular actions mainly via LPARs, on NB cell proliferation and migration with or without LPAR1/LPAR3 inhibitor Ki16425. The expression levels of LPAR1 and LPAR3 were detected, and the results showed that SH-SY5Y, SK-N-BE2 and CHLA-255 cells expressed LPAR1 but barely expressed LPAR3, indicating the main inhibitory effect of Ki16425 was against LPAR1 (Appendix A). Using three NB cell lines SH-SY5Y, SK-N-BE2 and CHLA-255, the proliferation was assessed by CCK-8 assays, while the migration was assessed either using Transwell or wound-healing assays. As shown in Figure 5A, LPA treatment with or without Ki16425 showed no effect on the proliferation of SH-SY5Y cells. In contrast, the decreased migrated cells of the LPA treatment group in Transwell assays and retarded wound closures of the LPA treatment groups in wound-healing assays (especially under the condition of unaffected proliferation) (Figure 5B,C) suggested that LPA could significantly inhibit the migration of SH-SY5Y cells. While Ki16425 exhibited no effects alone, LPAR1 inhibitor Ki16425 treatment abolished the inhibitory effects of LPA, suggesting the indispensable role of LPAR1 in the migration-inhibitory function of LPA. We also performed the same assays in other NB cells, SK-N-BE2 and CHLA-255, and obtained consistent results with those for SH-SY5Y (Figure 5D–I). The above results indicated that LPA suppressed the migration of NB cells via LPAR1.

### 3.6. Knockdown of LPAR1 Promoted the Migration of NB Cells

Subsequently, we knocked down LPAR1 in NB cells to identify its function. The efficiency of siRNA was first examined in HEK293T cells, and LPAR1-siRNA1 with a high knockdown efficiency was screened out (Appendix A). The expression of LPAR1 decreased significantly in siRNA-transfected SH-SY5Y, SK-N-BE2 and CHLA-255 cells (Figure 6A,D,G). Using these cells, CCK-8 and Transwell assays were performed. Our results in Figure 6B,E,H show that knockdown of LPAR1 had no effect on NB cell proliferation. Of note, knockdown of LPAR1 could promote the migration of NB cells, and LPA treatment hardly reversed the migration-promoting effect (Figure 6C,F,I), which verifies the significant role of LPAR1 in LPA for mediating the migration-suppressing function.

## 4. Discussion

Using bioinformatics analysis, mRNA microarray data analysis and experimental verification, our study aimed to identify DEGs between NB and control cells to further our understanding of the pathogenesis of NB and potentially provide diagnostic biomarkers and therapeutic targets. *FN1*, *PIK3R5*, *LPAR**6* and *LPAR**1* were screened out via KEGG, GO and PPI network analysis, and the research mainly focused on exploring the expression and function of LPAR1. We verified the lower expression level of LPAR1 in NB cells and further demonstrated that the LPA-LPAR1 axis suppressed the migration of NB cells.

LPAR1 was reported to be closely associated with the PI3K-Akt signaling pathway and tumor development [28], supporting the DEG screening procedures in our study. Accumulated research has revealed the decreased expression of LPAR1 and its migration-inhibiting effects in tumors including prostate cancer, gastric cancer and pancreatic cancer, which is consistent with our results [7,12,13]. However, LPAR1 expression has also been reported to be significantly increased in other tumors, such as human hepatic cancer [29], osteosarcoma [30] and ovarian cancer [31], and to exert tumor-promoting effects directly or mediated by chemotherapy resistance [30,32]. The controversial research results about LPAR1 suggest its different signaling transduction pathways and functions in different types of tumor cells. Some clinical trials of LPAR1 antagonists in cancer therapy were conducted, though there were no therapeutic trials or positive results reported [6]. Besides the different expression levels and signaling transduction patterns of LPAR1 in different tumors, another significant reason for the controversy around LPAR1’s function or failed clinical trials of LPAR1 antagonists is the mutations of *LPAR1* in cancer tissues. A study on metastatic neuroblastoma revealed an accumulation of *de novo* mutations, including a mutation of *LAPR1*, and identified that cells expressing the *LPAR1* R163W mutant showed significantly increased motility [33]. Several missense mutations of *LPAR1* were also found in rat cancer tissues [34]. When inducing MMP-2 expression and cell migration [35,36], or failing to show LPA-induced cellular responses [34], these *LPAR1* mutations resulted in changes to LPAR1’s function.

*FN1*, *PIK3R5* and *LPAR6* were also screened out by our bioinformatics analysis. It was reported that downregulation of FN1 (fibronectin 1) had no significant effects on NB cell proliferation, but it partially blocked ATRA-induced inhibition of cell migration and invasion in NB cells [37]. However, FN1 expression, when analyzed in our study, was not closely related to NB tumor stages and did not show significantly lower levels in high-risk NB tumors or NB tumors leading to patients’ death. LPAR6, another member of the LPAR family, showed extremely low expression levels in both NB cells and non-malignant cells, suggesting its minor function in NB. PIK3R5 is the regulatory subunit of PI3Kγ responsible for phosphorylating membrane lipids to activate the Akt pathway, and it is involved in tumorigenesis and progression. Suppressing the expression of PIK3R5 by miRNAs resulted in the promotion of epithelial-mesenchymal transition and oncogenic autophagy by regulating the Akt-mTOR signaling pathway in tumor cells [38]. Since there is a close association of LPAR1 with the PI3K-Akt signaling pathway, LPAR1 may cooperate with PIK3R5 to exert tumor-suppressing effects, which needs further exploration. Given that it is easier to apply the LPA-LPARs axis to clinical therapeutics for NB than it is to apply PIK3R5, we mainly focused on the expression and function of LPAR1 in NB in our study. Manipulating the ligand LPA could be a potential approach to NB therapy according to the function of the LPA-LPAR1 axis in our study. However, it is very difficult to apply just a ligand-protein LPA clinically, especially if the mutation of *LPAR1* and the function of other LPARs in NB tumor cells remain unclear.

Recently, the heterogeneity of neuroblastoma cells was defined by super-enhancer-associated transcription factors, such as *MYCN* and *PHOX2B*, and different tumor-cell subpopulations showed different characteristics of tumor development and metastasis [39,40,41]. Exploring the expression patterns and functions of LPAR1 in different subpopulations will be necessary in further studies. Beyond this, the heterogeneity of neuroblastoma, especially of metastasis-related changes in the bone marrow environment, was identified by RNA-sequencing analysis and single-cell analysis. A study revealed great diversity among disseminated NB tumor cells, and suggested that FAIM2 (Fas apoptotic inhibitory molecule 2) might be a complementary marker to capture metastatic tumor cells [42]. Beyond that study and our analysis based on data from the bone marrow of NB patients, the expression and function of LPAR1 in metastatic NB tumor cells remain to be further explored. Whether its expression is heterogeneous in particular subpopulations, and whether a particular subpopulation with extremely low LPAR1 expression plays the determining role in chemotherapy/radiotherapy resistance, are worthwhile investigating. 

## 5. Conclusions

Taken together, our findings demonstrate the downregulation of LPAR1 in NB cells and the tumor-suppressing effects of the LPA-LPAR1 axis. We suggest that LPAR1 may represent a potential target for future treatment of NB.

## Figures and Tables

**Figure 1 cancers-14-03346-f001:**
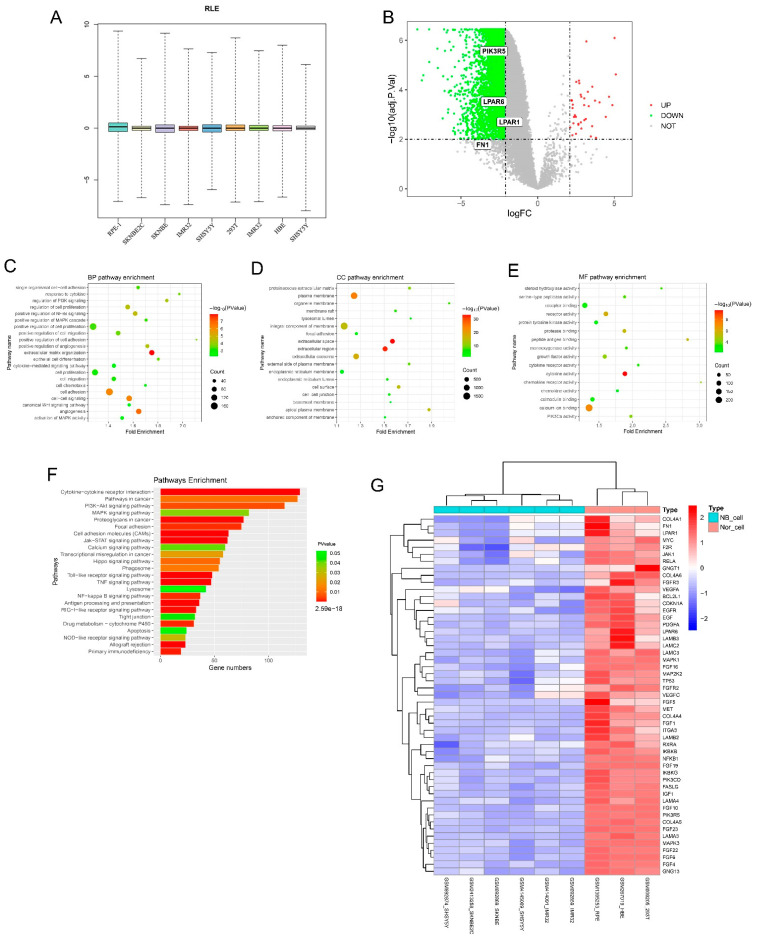
Identification of DEGs using mRNA microarray data analysis and GO/KEGG enrichment analysis. (**A**) Boxplots of RLE indicate the normalized raw data of microarray gene expression datasets. (**B**) Volcano plot distribution of all DEGs, with red points for the screened upregulated DEGs and green points for the screened downregulated DEGs. (**C**–**E**) Bubble chart visualization for GO analysis of all DGEs in NB cells and non-malignant cells. GO BP analysis (**C**), GO CC analysis (**D**) and GO MF analysis (**E**). (**F**) KEGG pathway analysis of unique DEGs in NB cells and non-malignant cells. (**G**) Hierarchical clustering analysis (heatmap) of 51 DEGs overlapping between PI3K-Akt pathways and pathways in cancer.

**Figure 2 cancers-14-03346-f002:**
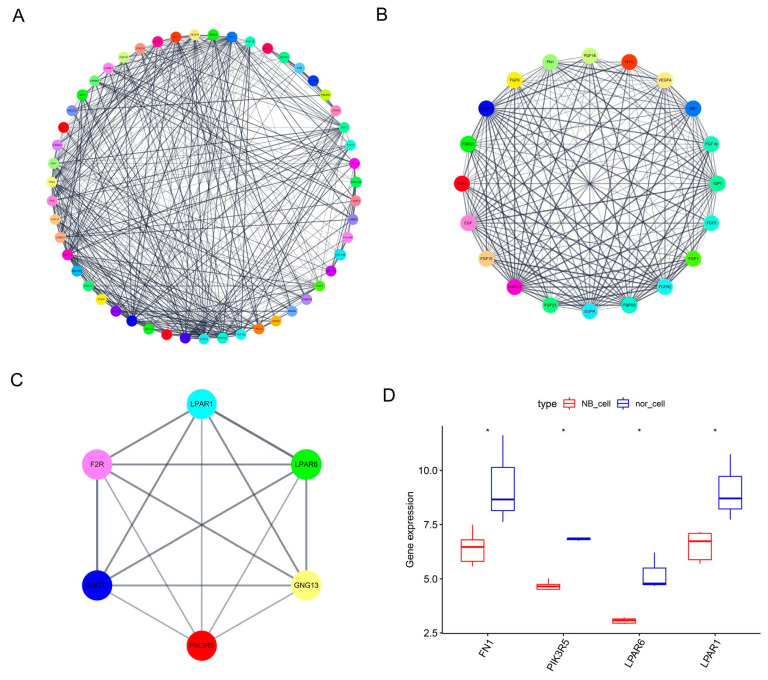
PPI network construction, module analysis and hub gene determination. (**A**) PPI network of screened genes was analyzed using STRING and Cytoscape for visualization. (**B**,**C**) Hub genes of protein interaction networks selected using MCODE. (**D**) Boxplot analysis was performed to identify the decreased expression of *FN1*, *PIK3R5*, *LPAR6* and *LPAR1*, with a high degree of network connectivity in the NB cells compared to the non-malignant cells. * *p* < 0.05.

**Figure 3 cancers-14-03346-f003:**
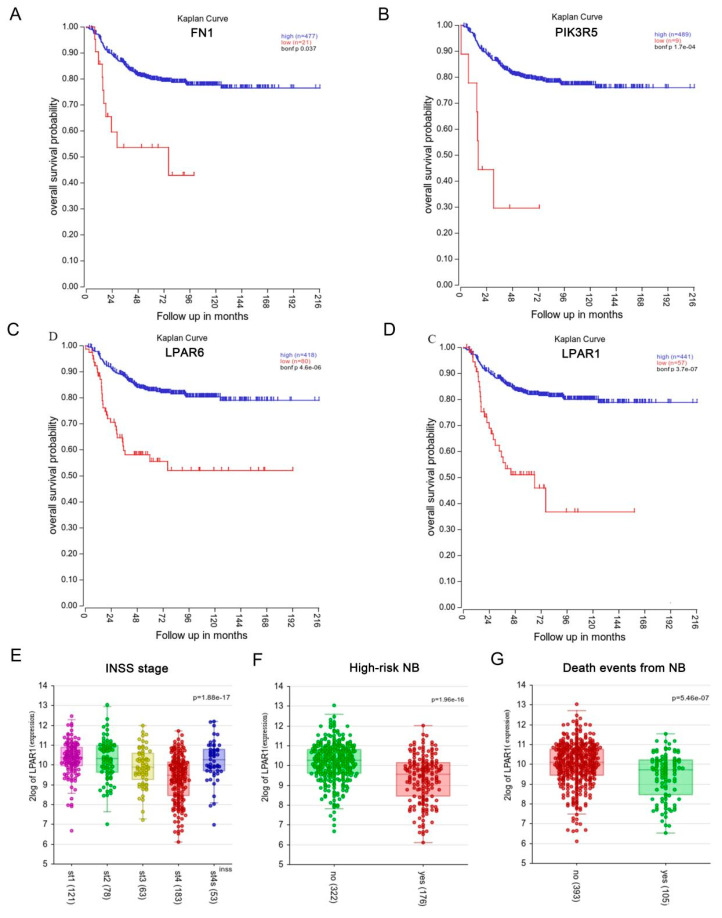
Hub gene expression and survival analysis. (**A**–**D**) Kaplan–Meier survival analysis for the SEQC datasets of 498 NB patients based on the average mRNA expression. Survival curves of FN1 (**A**), PIK3R5 (**B**), LPAR6 (**C**) and LPAR1 (**D**) in NB are shown, where *p* < 0.05 is regarded as the critical point with statistical significance. (**E**–**G**) R2 database view-a-gene was used to analyze the association between the LPAR1 expression and the NB INSS stage (**E**), likelihood of being high-risk (**F**) and likelihood of a death event (**G**) based on the average mRNA expression of the 498 NB SEQC datasets, with *p* < 0.05 regarded as the critical point with statistical significance.

**Figure 4 cancers-14-03346-f004:**
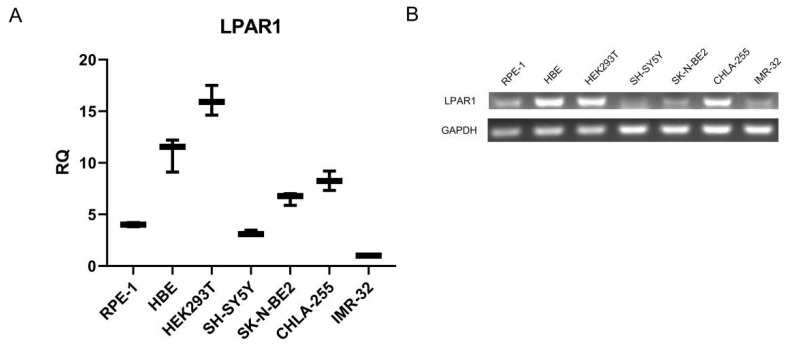
NB cells showed low expression level of LPAR1 compared to non-malignant cell lines. (**A**,**B**) The expression of LPAR1 at the mRNA level was analyzed by real-time PCR (**A**) and PCR (**B**) in NB cells and non-malignant cells. Original blots see Appendix A.

**Figure 5 cancers-14-03346-f005:**
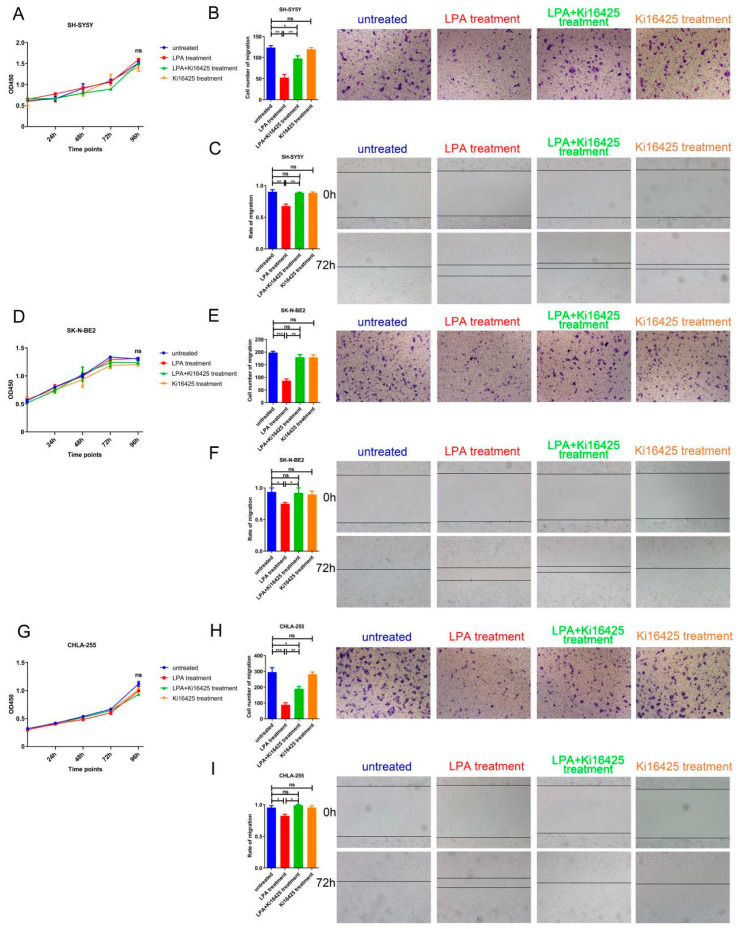
LPA suppressed the migration of NB cells via LPAR1. (**A**,**D**,**G**) CCK-8 assays were performed using SH-SY5Y, SK-N-BE2 and CHLA-255 cells treated with 10 μM LPA, LPA plus 10 μM Ki16425 or Ki16425 alone. (**B**,**E**,**H**) Transwell assays were performed using SH-SY5Y, SK-N-BE2 and CHLA-255 cells treated with 10 μM LPA, LPA plus 10 μM Ki16425 in the upper chamber or Ki16425 alone. Representative images of migrated cells obtained from the Transwell (magnification ×200) are shown (**right**). The cell numbers obtained from the Transwell assays were counted (**left**). (**C**,**F**,**I**) Wound-healing assays were performed and representative images (magnification ×100) are shown (**right**). The relative migration rate obtained from the wound-healing assays was calculated by dividing the change in the distance between the scratch edges by the initial distance (**left**). The results are expressed as the means ± SEMs from three independent experiments conducted in triplicate. * *p* < 0.05, ** *p* < 0.01 and *** *p* < 0.001 compared to the controls.

**Figure 6 cancers-14-03346-f006:**
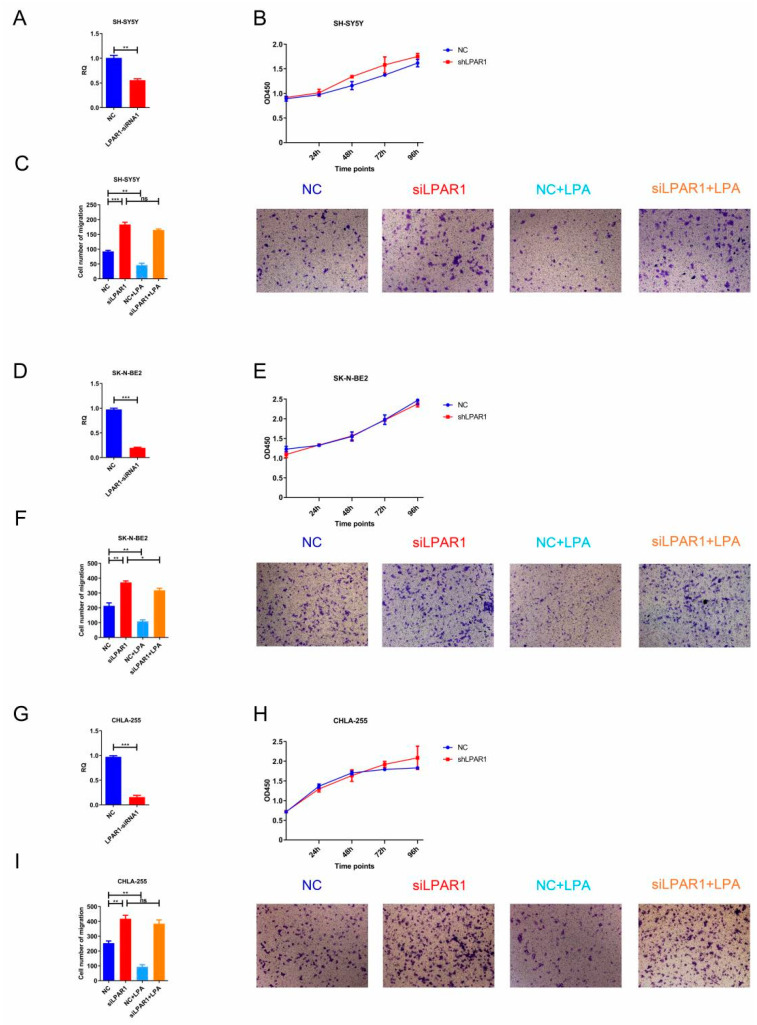
Knockdown of LPAR1 promoted the migration of NB cells. (**A**,**D**,**G**) The LPAR1 knockdown efficiency was analyzed by real-time PCR. (**B**,**E**,**H**) CCK-8 assays were performed using SH-SY5Y, SK-N-BE2 and CHLA-255 control cells and LPAR1 knockdown cells treated with 10 μM LPA. (**C**,**F**,**I**) Transwell assays were performed using SH-SY5Y, SK-N-BE2 and CHLA-255 control cells and LPAR1 knockdown cells treated with 10 μM LPA in the upper chamber. Representative images of migrated cells obtained from the Transwell (magnification ×200) are shown (**right**). The cell numbers obtained from the Transwell assays were counted (**left**). The results are expressed as the means ± SEMs from three independent experiments conducted in triplicate. * *p* < 0.05, ** *p* < 0.01 and *** *p* < 0.001 compared to the controls.

**Table 1 cancers-14-03346-t001:** GO analysis of DEGs.

Category	GO ID	Term	Gene ID	*p*-Value
BP	GO:0030198	Extracellular matrix organization	VIT, ITGB4, TNC, F11R, TNF, DAG1	1.08 × 10^−8^
BP	GO:0001525	Angiogenesis	CIB1, CTGF, LEPR, SYK, EREG, TGFA	1.22 × 10^−7^
BP	GO:0007155	Cell adhesion	TNC, COMP, TNR, FEZ1, CD151, LPP	2.70 × 10^−6^
BP	GO:0043123	Positive regulation of NF-κB signaling	TNF, CCR7, LTF, IRF3, LPAR1, NEK6	1.89 × 10^−5^
BP	GO:0042127	Regulation of cell proliferation	TES, CXCL3, JAK1, FGR, ACE2, LCK	1.06 × 10^−5^
BP	GO:0014066	Regulation of PI3K signaling	KLB, EGFR, IER3, BTC, NRG4, IRS1	5.35 × 10^−5^
BP	GO:0030335	Positive regulation of cell migration	ILK, PLAU, FGR, LEF1, CCL7, DAB2	3.02 × 10^−4^
BP	GO:0000187	Activation of MAPK activity	LPAR1, PLCE1, GRM4, WNT5A, MOS	0.003828
CC	GO:0005886	Plasma membrane	SLA2, LIPH, AR, ACE2, FPR3, MYO6	2.53 × 10^−25^
CC	GO:0009986	Cell surface	LIPG, KRT4, BST2, TF, CALR, SHH	1.58 × 10^−4^
CC	GO:0005578	Proteinaceous extracellular matrix	GLDN, TNR, LOX, PI3, CILP, CALR	1.66 × 10^−10^
CC	GO:0031090	Organelle membrane	FAAH, TFPI, FMO1, FA2H, CYP2S1	3.28 × 10^−7^
CC	GO:0005911	Cell-cell junction	MLC1, KRT8, DSG2, TLN1, VCL	9.37 × 10^−5^
CC	GO:0005925	Focal adhesion	TNC, PVR, TNS4, EZR, PXN, CALR	0.002148
CC	GO:0005789	Endoplasmic reticulum membrane	ALG1, POR, HPD, RCE1, PIGS, PIGZ	0.008747
MF	GO:0005509	Calcium ion binding	SYTL2, REG4, AIF1L, EHD1, CALR	2.32 × 10^−6^
MF	GO:0004872	Receptor activity	PVR, THBD, TLR1, LRP1, CALCR	1.06 × 10^−6^
MF	GO:0046934	PIK3Ca activity	KLB, PIK3R5, EGF, BTC, LCK, NRG4	7.22 × 10^−5^
MF	GO:0004896	Cytokine receptor activity	FLT3, MPL, CSF2RB, OSMR, CD44	0.001221

**Table 2 cancers-14-03346-t002:** KEGG pathway analysis of DEGs.

KEGG ID	Term	Gene ID	*p*-Value
hsa04060	Cytokine-cytokine receptor interaction	MPL, EDAR, NGFR, LIF, EDA, PRL	2.59 × 10^−18^
hsa04514	Cell adhesion molecules (CAMs)	PVR, SPN, CTLA4, CD8A, SELP	1.01 × 10^−5^
hsa04630	Jak-STAT signaling pathway	OXTR, LEPR, LPAR1, MC2R, PLG	4.98 × 10^−5^
hsa04668	TNF signaling pathway	RELA, JUN, EDN1, JAG1, MLKL	2.38 × 10^−6^
hsa04064	NF-kappa B signaling pathway	PTGS2, RELA, PLAU, SYK, LTBR	0.002268
hsa04510	Focal adhesion	MYLK, TNR, VWF, VCL, SRC, SPP1	0.002427
hsa04151	PI3K-Akt signaling pathway	TP53, LPAR1, CHAD, PCK1, PRL	0.005575
hsa05200	Pathways in cancer	MITF, TP53, LPAR1, LPAR6, FLT3	0.011031
hsa04010	MAPK signaling pathway	FOS, TP53, RRAS, FLNB, NTRK2	0.037591
hsa04020	Calcium signaling pathway	RYR1, OXTR, PLCE1, ORAI1, ITPR3	0.040378

## Data Availability

The data presented in this study is available on request from the corresponding author.

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
