# Peer review of "Reduction of LPAR1 Expression in Neuroblastoma Promotes Tumor Cell Migration"

_cancers, 2022, doi:10.3390/cancers14143346_

Round 1

Reviewer 1 Report

In this manuscript the authors present simple and clear analyses of both in silico and in vitro data of Neuroblastoma cell lines. However, translation of these findings to clinical tumors is problematic.

The starting point (3.1 and 3.3) is a bioinformatics analysis of neuroblastoma cell lines versus “normal cells”. There are a few mayor issues with this. 1) these cell lines have been in culture for decades and are not representative of the neuroblastoma tumors. More recently establish lines/organoids/TICs are available and are shown to cluster closer to neuroblastoma tumors. 2) Why not just use tumor data available for these in silico analyses? 3) The analysis should be done in comparison to “normal cells” from the same lineage (neural crest or adrenal) not some randomly selected lines such as HEK. 4) Why focus on these down-regulated genes? If you are looking for a potential druggable target, upregulated genes are more logical.

3.2 is missing.

3.4 does show a possible in vivo correlation, however an important analysis is missing. Since most neuroblastoma deaths (and tumor metastasis) occur in the high-risk group (inss stage 4), the correlation with LPAR1 expression should be analyzed separately in only these tumors.

3.5 These are solid experiments to evaluate the correct in vitro tools / cell lines to perform any further assays. But an important remark to be made is that neuroblastoma cell lines in general have a much lower LPAR1 expression (MASS5.0; easily checked on R2 platform) than neuroblastoma tumors. In fact, the expression in neuroblastoma tumor is much higher than in many other tumors such as lung, colon, breast etc..

3.6 These experiment are lacking the Ki16425 only control, and should be done with low serum conditions to eliminate proliferation effects.

3.7 Nice experiment in vitro, but difficult to translate to a clinical perspective. Do the authors suggest to upregulate LPAR1 in neuroblastoma tumors as a therapy? As noted before; LPAR1 expression is already much higher in NB tumors compared to cell lines.

Additionally, recent publications in multiple high-impact journals demonstrate neuroblastoma tumors to be consisting of different cell types/states with different properties. One of these sub-populations is very much linked to migration and metastasis. The discussion should address the implication of this data in those findings crucial to the field of neuroblastoma (treatment).

Reviewer 2 Report

This manuscript shows a well-designed and robust investigation of new genes involved in NB progression, providing new potential therapeutic targets. The valid bioinformatics analysis applied on publicly available datasets, led authors to select relevant genes (FN1, PIK3R5, LPAR6 and LPAR1) associated with NB patients survival and specifically over expressed in HR-NB patients. The low expression of such genes was correlated to poor clinical outcome and authors confirmed their down regulation in NB cell lines compared to non-malignant cells. The manuscript focused on the functional analysis of LPAR1 gene, which was shown to be involved in the inhibition of cell migration by performing its siRNA-mediated knock-down. Given the role of LPAR1 in NB cell migration, authors suggest that this gene may be a new therapeutic target for NB treatment. 

I have few comments for the authors:

1) In supplementary figure 1A, the box plots showing FN1 expression in HR vs non HR-NB patients and in alive vs deceased patients are not reported. Why authors did not perform the same analysis? Was it not significant?

2) Considering that FN1, PIK3R5, LPAR6 and LPAR1 all showed significant association with NB patient survival, it is not clear why authors focused only on LPAR1 gene. Authors should better point out the reasons of their choice in the Results and Discussion sections.

3) In the Method section the procedure for the analysis of Real-time PCR is not explained. Please, can author add more details? 

4) In figure 4A, the RQ shown on the Y axis refers to the data reported as 2^(-DCT)? This should be specified. Are the results derived from 3 replicates of the expression measurement? If this is the case, may I suggest to use box-plots instead of bar-chart, and report the significance of the difference observed between non malignant and NB cells?

5) At which time-point the RTqPCR for LPAR1 expression after siRNA transfection (Figure 6) was performed? Did authors check for the stability of siRNA efficiency over time? 

6) Why authors tested the LPAR1/LPAR3 inhibitor Ki16425 and LPAR1 knockdown in all the NB cell lines analyzed for LPAR1 expression except IMR32 cells?

Round 2

Reviewer 1 Report

Response Q1) Analysis on this dataset of bone marrow samples is not very representative for NB tumors either; it’s unclear how many infiltrating tumor cells are in these samples and the data mainly reflects expression of hematopoietic as indicated in the last sentence of the description of GSE25624.

Response Q2+3) Most new relevant expression data on NB available and described is scRNA-seq data; for instance from:

Dong R, Yang R, Zhan Y, Lai H-D, Ye C-J, Yao X-Y, et al. Single-Cell Characterization of Malignant Phenotypes and Developmental Trajectories of Adrenal Neuroblastoma. Cancer Cell 2020;38(5):716-33.
Also see BioRxiv manuscripts (below) for more datasets.

Response Q4) Maybe a more riveting target, but not practically applicable.

Response 3.4) “Another possibility is that within the state 4 group, there is no strict subgroup

classification based on clear demarcation of severity of the disease” à you will find subgroups with many other genes though.

Response 3.5) see response Q1

Response 3.6) In wound healing assays it is preferred to eliminate/reduce proliferation; especially when looking at relative long time-periods as 72hr.

Additionally, I’m not only referring to recent publication of metastasis, but also of the metastatic potent of NB tumor cells. For instance:

# van Groningen T, Koster J, Valentijn LJ, Zwijnenburg DA, Akogul N, Hasselt NE, et al. Neuroblastoma is composed of two super-enhancer-associated differentiation states. Nat Genet 2017;49(8)

# Boeva V, Louis-Brennetot C, Peltier A, Durand S, Pierre-Eugene C, Raynal V, et al. Heterogeneity of neuroblastoma cell identity defined by transcriptional circuitries. Nat Genet 2017;49(9):

# Gartlgruber M, Sharma AK, Quintero A, Dreidax D, Jansky S, Park Y-G, et al. Super enhancers define regulatory subtypes and cell identity in neuroblastoma. Nature Cancer 2021;2(1):

# Jansky S, Sharma AK, Körber V, Quintero A, Toprak UH, Wecht EM, et al. Single-cell transcriptomic analyses provide insights into the developmental origins of neuroblastoma. Nature Genetics 2021;53(5)

# Yuan X, Seneviratne JA, Du S, Xu Y, Chen Y, Jin Q, et al. Single-cell RNA-sequencing of peripheral neuroblastic tumors reveals an aggressive transitional cell state at the junction of an adrenergic-mesenchymal transdifferentiation trajectory. bioRxiv 2020:2020.05.15.097469 doi 10.1101/2020.05.15.097469.

# Wolpaw AJ, Grossmann LD, Dong MM, Dessau JL, Brafford PA, Volgina D, et al. Epigenetic state determines inflammatory sensing in neuroblastoma. bioRxiv 2021:2021.01.27.428523 doi 10.1101/2021.01.27.428523.

# Olsen TK, Otte J, Mei S, Kameneva P, Björklund Å, Kryukov E, et al. Malignant Schwann cell precursors mediate intratumoral plasticity in human neuroblastoma. bioRxiv 2020:2020.05.04.077057 doi 10.1101/2020.05.04.077057.
